# Machine learning prediction of weight gain after antiretroviral therapy initiation in people with HIV: Insights from a large french real-world cohort

Cyrielle Codde[1,2]*, Clément Benoist[2], Laurent Hocqueloux[3], Cyrille Delpierre[4], Clotilde Allevena[5], Amélie Ménard[6], Antoine Chéret[7], Cédric Arvieux[8], Jean-François Faucher[1], Jean-Baptiste Woillard[2,9], on behalf of the Dat'AIDS Study Group[¶]

**1** Department of infectious diseases, Dupuytren 2 University Hospital, Limoges, France, **2** Inserm Pharmacology & Toxicology, U 1248, F-87000, Limoges University, Dupuytren University Hospital, Limoges, France, **3** Department of infectious diseases, Orléans University Hospital, Orléans, France, **4** Toulouse University, Toulouse, France, **5** Department of infectious diseases, Nantes University Hospital, Nantes, France, **6** Department of infectious diseases, Marseille University Hospital Instutit, Marseille, France, **7** Department of infectious diseases, Kremlin Bicêtre University Hospital, Paris, France, **8** Department of infectious diseases, Rennes University Hospital, Rennes, France, **9** Department of Pharmacology, Toxicology, and Pharmacovigilance, Dupuytren University Hospital, Limoges, France

¶ Membership of the Dat'AIDS Study Group is provided in the Acknowledgements.
☯ These authors contributed equally to this work.
* ccodde@yahoo.com, cyrielle.codde@chu-limoges.fr

## Abstract

Excessive weight gain after initiation of antiretroviral therapy (ART) has become a recognized concern among people living with HIV. Individual weight trajectories remain highly heterogeneous and challenging to predict using conventional methods. We leveraged the French Dat'AIDS national cohort to assess whether machine learning (ML) could enhance the prediction of individual body weight at 6, 12, and 24 months after ART initiation. Using 112 baseline variables encompassing demographic, clinical, laboratory, and treatment-related data, we trained XGBoost models and evaluated performance using root mean square error (RMSE), $R^2$, and mean prediction error. A simple benchmark model based on baseline weight was used for comparison. Among 24,014 eligible ART-naïve adults, the ML models achieved RMSEs of approximately 4.6 kg, 5.3 kg, and 6.4 kg at 6, 12, and 24 months respectively, with declining predictive power over time. Baseline weight (Weight_M0) consistently emerged as the strongest predictor, while other factors contributed minimally. Although ML marginally outperformed the benchmark (Weight_M0), accuracy remained insufficient for clinical decision-making. Sensitivity analyses excluding individuals with implausibly large monthly weight changes modestly improved RMSE (3.9–6.0 kg), underscoring the impact of data quality. Our results demonstrate that, despite large sample size and rich clinical variables, ML lacks the precision necessary for individual weight forecasting in this context. These findings highlight the limitations of applying artificial intelligence to heterogeneous real-world cohorts and

**Data availability statement:** The data underlying this study are drawn from the French Dat'AIDS cohort. These data cannot be shared publicly due to national data protection regulations (Commission Nationale de l'Informatique et des Libertés, CNIL). Access to Dat'AIDS data may be granted upon reasonable request to the Dat'AIDS scientific committee (president: Laurent Hocqueloux; laurent.hocqueloux@chu-orleans.fr and data protection officer: dpo@dataids.com), subject to compliance with French regulations and institutional agreements.

**Funding:** The author(s) received no specific funding for this work.

**Competing interests:** The authors have declared that no competing interests exist.

underscore the need to incorporate behavioral and lifestyle factors to improve predictive modeling.

## Introduction

Excessive weight gain following initiation of antiretroviral therapy (ART) is increasingly recognized in people living with HIV (PLWH) [1,2]. This phenomenon is particularly associated with integrase strand transfer inhibitors (INSTIs) and tenofovir alafenamide (TAF) [3–5], but weight trajectories remain highly heterogeneous [6,7]. While weight gain may reflect a desirable "return-to-health" in patients presenting with HIV-related wasting, it can also lead to overweight, obesity, and cardiometabolic complications in those with normal baseline weight.

Several population-based studies have identified risk factors for weight gain, including female sex, Black ethnicity, and exposure to specific antiretroviral drugs [3,8–10]. However, individual prediction remains challenging [4]. Traditional linear regression approaches have provided insights at the group level but fail to capture the complexity of weight evolution at the patient level, which depends on medical, behavioral, and lifestyle factors that are not always measured.

Machine learning (ML) approaches are well suited to model complex, nonlinear relationships and may therefore improve individual-level prediction. In other fields, ML has been successfully applied to forecast weight gain during pregnancy or after exposure to psychotropic drugs [11]. In HIV research, ML has mainly been used to predict virological suppression, treatment adherence, or resistance, but little is known about its capacity to predict weight trajectories under ART [12].

Recent pilot studies, such as the work by Motta et al.[13], have explored ML algorithms in specialized cohorts with moderate success. However, a critical research gap remains: the scalability of these models to large, unselected real-world populations. It is currently unknown whether the predictive signals identified in smaller, homogeneous datasets remain robust when applied to nationwide heterogeneous cohorts, or if they are diluted by the variability of routine clinical care.

The objective of this study was to develop and evaluate machine learning models for predicting individual weight trajectories at 6, 12, and 24 months after ART initiation in treatment-naïve PLWH from the French Dat'AIDS cohort. Beyond assessing prediction accuracy, we aimed to highlight the opportunities and limitations of applying machine learning to large heterogeneous real-world datasets.

## Methods

### Study design and population

We conducted a retrospective cohort study using data from the French Dat'AIDS cohort, a nationwide collaboration of 16 HIV reference centers. Dat'AIDS collects standardized longitudinal clinical information from PLWH receiving routine care.

We included all ART-naïve adults (≥18 years) who initiated combination ART between 01/01/2004 and 31/12/2021 and had at least one available weight

measurement at baseline and during follow-up. Patients with missing baseline weight or incomplete ART regimen data were excluded. Data were accessed for research purposes from 01/03/2023 to 31/10/2023.

## Outcome

The primary outcome of interest was body weight, expressed in kilograms, at 6, 12, and 24 months after ART initiation. When multiple weight measurements were available within a time window of ±2 months around the target visit, the measurement closest to the corresponding timepoint was retained.

## Predictors

A total of 112 baseline variables were considered as potential predictors after multidisciplinary discussion (pharmacologist, infectious disease specialist, data scientist) and are presented in Table 1.

**Table 1. Predictors selected for machine learning analysis of weight gain in PWH.**

| Socio-Demographic and Follow-up Predictors | | | |
|---|---|---|---|
| COREVIH center number<br>Gender<br>Time interval between visit and HIV diagnosis | Weight at first visit<br>Birth country<br>Age at visit n | | Weight at visit n<br>Year and month at visit n |
| **Biological predictors** | | | |
| Log HIV viral load at visit n<br>CD4 count at visit n, *values bounded between 200–3000 (/mm3)*<br>CD4 CD8 ratio at visit n, *values bounded between 0–4* | | | |
| **Predictors related to HIV history** | | | |
| Year of HIV diagnosis<br>HIV type (1, 2)<br>AIDS stage at visit n<br>Time since initiation of ARV line | Second-generation INSTI†<br>TAF†<br>INSTI | | NRTI<br>NNRTI<br>PI |
| Raltegravir<br>Elvitegravir<br>Dolutegravir<br>Bictegravir<br>Cabotegravir<br>Saquinavir<br>Ritonavir | Lopinavir<br>Atazanavir<br>Fosamprenavir<br>Tipranavir<br>Darunavir<br>Indinavir<br>Zidovudine | Lamivudine<br>Abacavir<br>Tenofovir disoproxil fumarate<br>Emtricitabine<br>Stavudine<br>Didanosine | Nevirapine<br>Efavirenz<br>Etravirine<br>Rilpivirine<br>Doravirine<br>Maraviroc<br>Islatravir<br>Bms955176 |
| **Predictors related to comorbidities** | | | |
| HBV, HCV co-infection<br>Metabolic disorders†<br>Endocrine risk factors†<br>Pregnancy†<br>Time since pregnancy<br>Menopause†<br>Time since menopause | Sedentary lifestyle, mobility restriction†<br>Diet and hygiene habits†<br>Socio-professional risk factors†<br>Malnutrition‡<br>Thyrotoxicosis‡<br>Anorexia‡<br>Intake of toxic substances‡ | | Mood disorders§<br>Schizophrenia disorders§<br>Bulimia§<br>Sleep disorders§<br>Non-compliance with treatment§<br>Enterocolitis§<br>Stomies§ |
| **Predictors related to co-medications** | | | |
| Atypical neuroleptics†<br>Tricyclic antidepressants†<br>Other antidepressants† | Thymoregulators and anti-convulsants†<br>Corticosteroids† | | GLP1 analogues‡<br>Medications associated with weight loss (others) ‡<br>Time since each medication |

INSTI, Integrase strand transfer inhibitor; TAF, Tenofovir alafenamide; NRTI, Nuclos(t)idic reverse transcriptase inhibitor; NNRTI, Non nuclos(t)idic reverse transcriptase inhibitor; PI, Protease inhibitor.

† Supposed predictors of weight gain, ‡ supposed predictors of weight loss, § supposed predictors of weight variation.

These included demographic characteristics such as age, sex, and geographical origin; clinical parameters such as baseline weight, body mass index (BMI), HIV disease stage, and comorbidities including hypertension, diabetes, renal or hepatic disease; laboratory results including CD4 and CD8 cell counts, plasma HIV RNA, creatinine, and liver function tests; and treatment-related variables such as ART regimen composition by drug class and individual antiretroviral agents. Lifestyle factors such as smoking, alcohol use, and history of opportunistic infections were also incorporated. For women aged 50 years or older, menopausal status was imputed based on age, which we acknowledge as an approximation with potential limitations.

The selection of comorbidities was based on the International Code of Diseases 10 (ICD-10), and the selection of comedications of interest and ARV treatment lines were identified according to their International Nonproprietary Names (INNs) and marketed specialty names. These selections are presented in the S1 Table.

Unlike the starting date of a comorbidity or comedication, the date of stop was not known in the Dat'AIDS database. Therefore, each antecedent or treatment of interest entered only once was considered continuous during longitudinal follow-up, with the exception of pregnancy, which was terminated after 9 months. Weights corresponding to entries when CD4 was less than 200/mm3 were omitted to restrict weight gain to that linked with a return to health rather than exclusively with ARV treatment exposure.

## Data preprocessing

Predictors with more than 30% missing values were excluded from the analysis. Data processing, imputation of missing data, correction of outliers, and graphical exploration were performed using the tidyverse and tidymodels packages [14,15]. Imputation was performed using knn approach. Continuous biological variables were bounded to eliminate outliers, resulting in a Gaussian distribution. Continuous variables were standardized, and categorical variables were transformed using one-hot encoding.

For all analyses, data cleaning consisted in restricting the weight values reported between 30 and 230 kg, in order to obtain a Gaussian distribution of outcome values with a maximum variation in weight per month lower or equal to 10 kg. Sensitivity analyses were performed by framing this maximum variation at 5 kg and 3 kg.

## Machine learning modeling

We applied the XGBoost (Extreme Gradient Boosting) algorithm to predict weight at 6, 12, and 24 months after ART initiation. Data splitting was performed allocating 75% in the train set (5053, 4283, and 3549 patients respectively) and 25% in the test set (1686, 1430 and 1185 patients). XGBoost models were trained on each train set to predict the weight at the checkpoint of interest. Each train set was used to tune the hyperparameters (trees, tree depth, learning rate, min_n, loss_reduction, sample_size, and mtry) and to evaluate the model performances by 10-fold cross-validation. The specific hyperparameter settings for the final models are detailed in S2 Table. Each best model was then evaluated in the corresponding test set by measuring the root mean square error (RMSE; expressed in kg) between the predicted and reference weight at the checkpoint. The performances were evaluated by calculation of the RMSE, r2, mean prediction error (MPE; expressed in kg), relative MPE (%), and relative RMSE (%) in the test set. Variable importance plot obtained by random permutation were drawn to assess the importance of predictors. Finally, scatter plots of predicted vs reference weight at each checkpoint in the test set were drawn.

## Benchmark models

To contextualize the performance of the machine learning approach, we also developed a LASSO penalized multivariable Linear Regression model using the same set of predictors and data splitting strategy (training/testing sets) as the XGBoost model at 6, 12 and 24 months. This served to assess whether a non-linear approach provided a significant advantage over traditional statistical modeling.

 

In addition to the ML models, we defined a simple benchmark model based on baseline weight at ART initiation (Weight_M0). In this approach, the predicted weight at 6, 12, or 24 months was assumed to be equal to the baseline weight, i.e., assuming no change over time. For each timepoint, we calculated the same performance metrics as for the ML models.

## Sensitivity analyses

To account for potential measurement errors and implausible fluctuations in weight trajectories, we conducted sensitivity analyses by restricting the study population to individuals with limited monthly variations in body weight. Two alternative cohorts were defined: the first excluded patients with weight changes exceeding 5 kg per month, and the second excluded those with variations greater than 3 kg per month. For each restricted dataset, XGBoost models were re-trained and evaluated at 6, 12, and 24 months following ART initiation, using the same procedures for data preprocessing, model training, and performance assessment as in the main analysis.

## Ethics

This study was conducted in accordance with French ethics regulations and the database received approval from the Commission Nationale de l'Informatique et des Liberte´s (CNIL) and is registered in ClinicalTrials.gov. Medical records were collected from the Dat'AIDS cohort study, which is a collaboration of 30 HIV treatment centers in France and overseas (registered with ClinicalTrials.gov under the identifier NCT02898987). These centers maintain prospective cohorts of PLWH who provide written informed consent via a unique electronic medical record (Nadis®). Anonymized data for clinical events, laboratory test results and therapeutic history are collected by the networking organization, and the Dat'AIDS study was registered with the French National Commission on Informatics and Liberties (CNIL Registration number: 2001/762876/nadiscnil.doc). This study was carried out in compliance with the International guidelines for human research protection as per the Declaration of Helsinki and ICH-GCP.

# Results

## Study population

Dat'AIDS provided 11 separate tables: Data (socio-demographic data), Medical history (longitudinal entry of cohort comorbidities), Comedic (longitudinal entry of comedications), CVVIH (longitudinal monitoring of viral load), CD4 CD8 (longitudinal monitoring of CD4 and CD8 lymphocyte levels and ratio), Creat (longitudinal monitoring of serum creatinine), Leuco (longitudinal monitoring of leukocyte levels), Transa (longitudinal monitoring of transaminases), Lipids (longitudinal monitoring of the evaluation of lipid abnormalities), Exam_clinique (height, abdominal and hip circumference, blood pressure and weight), Evt_ther (ARV treatment line, with reason for stopping or switching). The number of patients and the overall percentage of missing data in each initial table are presented in the S3Table. After data cleaning and removal of outliers, we obtained a reduced number of patients for each of the tables.

Out of a theoretical number of 78,621 patients, 37,621 were diagnosed between January 1, 2004 and December 31, 2021 (DATA Table). Of this sample, 26,070 patients had at least 3 weights recorded in their follow-up (EXAM_CLINIQUE Table). Forty-seven of them had a history of ARV treatment according to the EVT_THER table, 2,009 patients did not have available biological values (CD4 CD8 and CVVIH tables) and were excluded. The flow chart is presented in Fig 1.

There were 24,014 patients who met the criteria for workable data. In this available population, 6,739, 5,713 and 4,734 patients treated in the first line, called naive, were analyzed for weight prediction models at 6, 12 and 24 months. The cohorts were on average about 40 years old, were made up of more than 70% men, and more than half were born in France. It was mainly HIV-1 infection with less than 15% of AIDS stage patients. At each checkpoints, there was an average weight gain of 2 kg in each cohort. Their characteristics are summarized in Table 2.

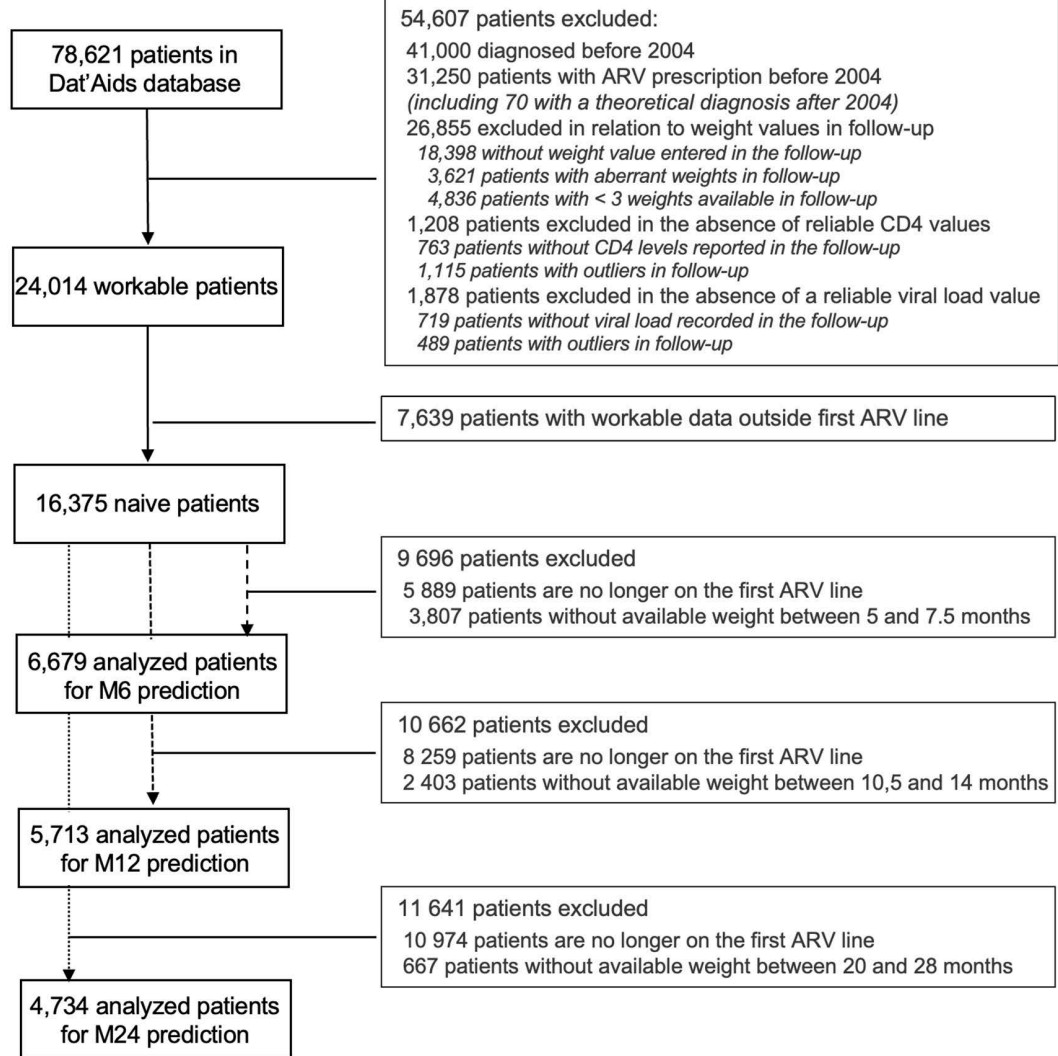

**Fig 1. Flow-chart.**

The characteristics for the subpopulations (sensitivity analyses) bounded by maximum monthly weight variations of 5 kg and 3 kg are presented in the S4 Table.

## Model performance

The results obtained after cross-validation (cutting into 10 subgroups) for each time checkpoint are presented in the Table 3.

They show fairly mediocre results with prediction inaccuracies in the test set at 4.60, 5.28 and 6.36 kg for the respective predictions at 6, 12 and 24 months. The calibration of the model was assessed visually using scatter plots of observed vs. predicted weights, presented in Fig 2. These plots illustrate the dispersion of predictions around the identity line.

The more we advance in time, the more the predictions are spread out agreeing with the different RMSE values.

**Table 2. Main characteristics of PWH treated in first line, so-called naive, used for weight prediction at 6, 12 and 24 months.**

| | Cohort M6 N = 6739 | Cohort M12 N = 5713 | Cohort M24 N = 4734 |
|---|---|---|---|
| Age (year)[1] | 40.27 (31.20-48.20) | 40.99 (32.10-49.00) | 42.06 (33.20-50.10) |
| Gender | Male: 4851 (71.98) | Male: 4190 (73.34) | Male: 3483 (73.36) |
| | Female: 1843 (27.35) | Female: 1484 (25.98) | Female: 1218 (25.73) |
| | Trans M-F: 45 (0.67) | Trans M-F: 39 (0.68) | Trans M-F: 33 (0.70) |
| Birth country | † | ‡ | § |
| *France* | 3759 (55.78) | 3285 (57.50) | 2720 (57.46) |
| *West and Central Africa* | 1403 (20.82) | 1149 (20.11) | 972 (20.53) |
| *Middle East and North Africa* | 244 (3.62) | 198 (3.47) | 182 (3.84) |
| *Others* | 789 (11.71) | 667 (11.68) | 576 (12.17) |
| VIH-1 | 6698 (99.39) | 5688 (99.56) | 4704 (99.37) |
| AIDS stage | 891 (13.22) | 727 (12.73) | 538 (11.36) |
| Weight at first visit (kg)[1] | 70.70 (62.00-78.00) | 71.02 (62.00-78.50) | 71.42 (62.50-79.00) |
| Weight at checkpoint (kg)[1] | 72.26 (63.00-80.00) | 73.16 (64.00-81.00) | 73.98 (64.00-82.00) |
| Viral load (log)[1] | 1.69 (1.30-1.68) | 1.67 (1.30-1.60) | 1.66 (1.30-1.60) |
| CD4 count (/mm3)[1] | 573 (379-711) | 606 (405-748) | 646 (446-794) |
| CD4 CD8 ratio[1] | 0.80 (0.46-1.02) | 0.87 (0.51-1.11) | 0.95 (0.58-1.21) |
| ARV treatment | | | |
| *TAF* | 593 (8.80) | 548 (9.59) | 463 (9.78) |
| *INSTI* | 1691 (25.09) | 1432 (25.07) | 1196 (25.26) |
| | *Including second generation INSTI: 878 (13.00)* | *Including second generation INSTI: 760 (13.30)* | *Including second generation INSTI: 679 (14.34)* |
| *NRTI* | 6572 (97.52) | 5584 (97.74) | 4628 (97.76) |
| *NNRTI* | 1469 (21.80) | 1393 (24.38) | 1353 (28.58) |
| *PI* | 3569 (52.96) | 2856 (49.99) | 2167 (45.78) |
| ≥ 1 Supposed comorbidities of weight gain | 1250 (18.55) | 1161 (20.32) | 1068 (22.56) |
| ≥ 1 Supposed comorbidities of weight loss | 126 (1.87) | 105 (1.84) | 106 (2.24) |
| ≥ 1 Supposed comorbidities of weight variation | 780 (11.57) | 761 (13.32) | 664 (14.03) |
| ≥ 1 Supposed comedications of weight gain | 204 (3.03) | 171 (2.99) | 175 (3.70) |
| ≥ 1 Supposed comedications of weight loss | 4 (0.06) | 2 (0.04) | 8 (0.17) |

Values in number (%), [1]Average (quartiles).

†Not reported: 544 (8.07), ‡Not reported: 414 (7.25), §Not reported: 287 (6.06).

ARV, Antireotroviral; INSTI, Integrase strand transfer inhibitor; TAF, Tenofovir alafenamide; NRTI, Nuclos(t)idic reverse transcriptaseinhibitor; NNRTI, Non nuclos(t)idic reverse transcriptaseinhibitor; PI, Protease inhibitor.

## Predictor importance

Baseline weight consistently emerged as the strongest predictor across all timepoints. Other variables such as age, sex, country of birth, CD4 cell count, and ART regimen contributed modestly to prediction. Lifestyle factors and comorbidities had limited influence on model performance (Fig 3).

**Table 3. Performance of XGBoost models for weight prediction in naive PWH at 6, 12 and 24 months in the test sets.**

|  | *Prediction M6* | *Prediction M12* | *Prediction M24* |
|---|---|---|---|
| RMSE, kg[a] | 4.60 | 5.28 | 6.36 |
| R2 of RMSE[a] | 0.893 | 0.862 | 0.808 |
| Relative RMSE, % | 6.21 | 6.88 | 8.27 |
| Relative biais, % | −0.08 | −0.10 | −0.87 |

[a] Value obtained after 10 cross-validation.

The RMSE evaluates the accuracy of the model by measuring the average difference between the actual values and the predictions (the lower it is, the better the performance of the model). The R2 of RMSE measures the quality of fit of the model in relation to the variability of the real data (0: no fit, 1: perfect fit). The relative RMSE expresses the relative error of the mean compared to the actual values. Relative bias measures the systematic error of the model compared to actual values.

### Benchmark linear model and baseline weight

The comparative LASSO penalized Linear Regression model yielded RMSE (relative RMSE/ relative biais) values in the test sets of 4,53 kg (6.23%/ 0.34%), 5.18 kg (6.85%/ 0,27%), and 6.18 kg (7.95%/ 0.72%) kg at 6, 12, and 24 months, respectively. The marginal difference between XGBoost and the linear model suggests that the predictive limitation lies in the informational content of the variables rather than the modeling technique.

When compared with the simple benchmark model assuming no weight change from baseline, XGBoost yielded lower RMSE values and higher R² values, indicating better fit. However, the magnitude of improvement was limited and insufficient for precise individual-level prediction (S5 Table).

### Sensitivity analysis: Bounded subpopulations on weight change per month

The sensitivity population were more restrictive for criteria and were reduced in number, with cohorts of 6,679, 5,670 and 4,726 patients for predictions with weight variations restricted at 5 kg per months at 6, 12 and 24 months respectively, and 6,540, 5,596 and 4,684 patients respectively for predictions with weight variations restricted at 3 kg per months. The XGBoost models developed from more strictly selected sub-populations showed improved performance, which would support the presence of outliers with aberrantly large weight variations. Once again, better metric values were obtained for prediction at 6 months than at 12 and 24 months. Thus, when the maximum variation in weight per month was capped at 5 kg and then 3 kg, we obtained RMSE values (S6 Table) of 4.42, 4.98 and 5.96 kg for predictions at 6, 12 and 24 months respectively, and 3.89, 4.82 and 5.93 kg for variation in weight per month capped at 3 kg. As for the main analysis, the weight at the first visit was the most important variable and when used of this value instead of the ML model prediction, the comparison with the metrics obtained using Weight_T0 were less efficient, thanks to our XGBoost models.

### Discussion

In this large national cohort of treatment-naïve people living with HIV, we evaluated the capacity of machine learning to predict individual weight evolution after ART initiation. Despite the use of more than one hundred baseline variables, a rigorous modeling strategy, and sensitivity analyses in carefully cleaned datasets, prediction accuracy remained limited, with RMSE values between 4 and 7 kg. This degree of error prevents the use of such models for clinically actionable individual prediction.

The advantages of ML compared to statistical methods classically used in population studies are numerous: it allows complex modeling of non-linear relationships between data, allowing the capture of more subtle and nuanced patterns, it also allows adaptation to the individual context for predictions personalized to each individual and their characteristics. The definite advantage of using XGBoost lies in its ability to handle a large amount of data. It can utilize datasets with correlated variables and is relatively resistant to overfitting, allowing for the consideration of a large number of variables.

(A)

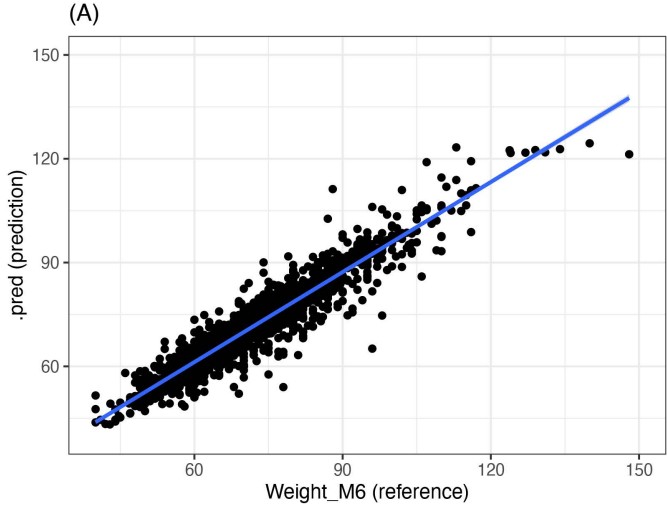

| Weight_M0 | Weight_M6 | .pred |
|---|---|---|
| 54 | 52 | 54.7 |
| 91 | 92 | 92 |
| 91 | 99 | 96 |
| 82 | 77 | 79.9 |
| 81 | 85 | 80.9 |

(B)

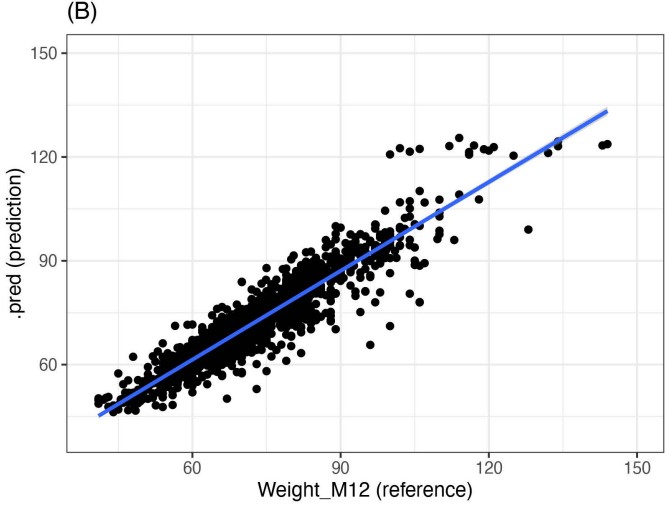

| Weight_M0 | Weight_M12 | .pred |
|---|---|---|
| 73 | 75 | 73.2 |
| 51 | 56 | 55.2 |
| 68 | 66.5 | 70.6 |
| 81 | 86 | 83.3 |
| 58 | 61 | 62.5 |

(C)

| Weight_M0 | Weight_M24 | .pred |
|---|---|---|
| 71 | 80 | 72.3 |
| 68 | 73 | 71.6 |
| 58 | 58 | 62 |
| 76 | 84.5 | 76.8 |
| 86 | 99 | 86.8 |

**Fig 2. Scatter plot of weights predicted by ML versus reference weight and examples of weight predictions at the checkpoint.** (A) 6-month prediction, (B) 12-month prediction, and (C) 24-month prediction.

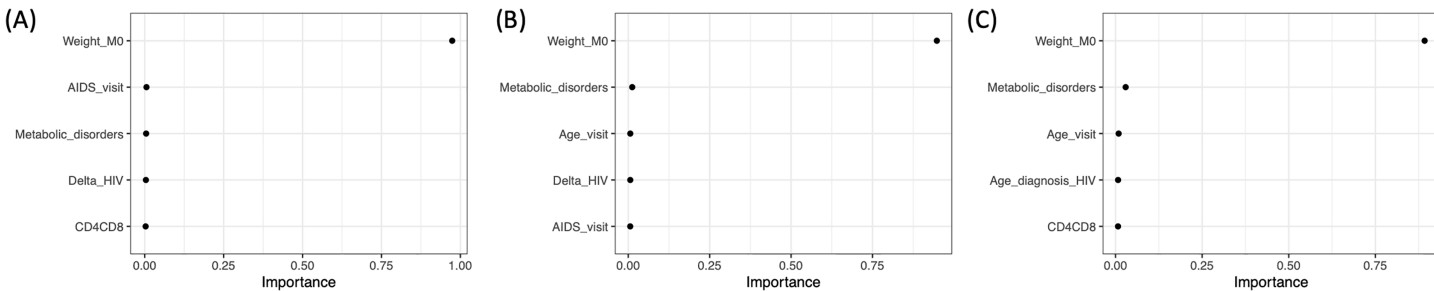

**Fig 3. Variable Importance Plot at 6 months (A), 12 months (B) and 24 months (C).**

The best results were obtained for predictions at 6 months, which is not surprising, the weight varies less overall at this time interval compared to longer periods of 12 or 24 months of exposure to ARV regimen. Furthermore, despite our efforts to group certain predictors together and thus limit the number of variables, we had more than a hundred. Informing as many parameters in the clinic, during an ARV treatment initiation consultation is not possible, unless an automatic extraction and preprocessing of the electronic health record. Even if the choice of predictors was made after multidisciplinary consultation, some of them presented obvious limitations: consideration of medical history and comedications was hampered by the absence of data concerning the duration of exposure to the predictor.

The number of patients available via the Dat'AIDS database was colossal, however, from a theoretical number of 78,621 patients, we were forced to considerably reduce the final number. Data cleaning was laborious, with missing data and many outliers to correct. The weight values, which were our key data in this project, were not systematically reported. Few patients ultimately had at least three "reliable" weights available for use in our models.

Finally, the most important predictors was baseline weight and would have preferred a distribution of the prediction balanced on sex, ethnicity, age, comorbidity, or target comedication, as found in population studies [3,8–10]. The consistent emergence of baseline weight as the primary predictor reflects the strong biological inertia of anthropometric measurements. In adults, body weight remains highly stable over time in the absence of acute illness or significant lifestyle modification. Consequently, baseline weight exerts a dominant statistical effect that overshadows the more modest contributions of ART regimens or immune parameters.

Several key lessons emerge from our study. First, the results highlight the limitations of real-world multicenter data for building clinically precise predictive tools. Although Dat'AIDS covers a nationwide population including a significant proportion of patients of non-European descent, the consistency of data reporting varies significantly across collection centers and individual practitioners. This heterogeneity is quantitatively illustrated by our flow chart: out of 16,375 eligible ART-naïve patients, only 6,679 had a recorded weight available for the 6-month prediction. This attrition rate underscores that weight is not systematically reported in the database during routine care, even if measured clinically. This variability creates a 'noisy' dataset where the presence of data often depends on provider habits rather than patient characteristics. Consequently, the model had to be trained on a selected fraction of the original cohort, limiting its accuracy and ability to generalize.

The improvement observed when excluding patients with implausibly large monthly weight variations illustrates how sensitive ML models are to data quality, reinforcing the fundamental principle of 'garbage in, garbage out. Thus, "more data" is not sufficient if the data are heterogeneous or noisy; standardized and high-quality phenotyping is essential.

Second, while we included a wide range of demographic, clinical, laboratory, and treatment-related variables, the prediction of weight trajectories is inherently influenced by behavioral and lifestyle determinants, such as dietary patterns, physical activity, and psychosocial factors, that were not available in our dataset. Their absence likely contributed to

the limited explanatory power of the models. To overcome these limitations, future predictive efforts must move beyond passive extraction of Electronic Medical Records. We recommend the integration of standardized Patient-Reported Outcomes (PROs) to actively capture determining factors such as dietary habits and psychosocial stressors. Furthermore, the increasing availability of digital health tools—such as connected activity trackers or mobile health applications—offers a way to collect granular, longitudinal data on lifestyle behaviors.

Our findings are consistent with those of Motta et al.,who evaluated weight prediction using machine learning in a smaller, specialized cohort and also reported limited precision (3.5–5 kg) [13]. Similarly, trajectory analyses from the US CNICS cohort [16] and pooled analyses of international randomized trials [5] have highlighted the extreme heterogeneity of weight gain, which remains largely unexplained by standard demographic and clinical variables. Taken together, these studies demonstrate that even in diverse healthcare settings (Europe, USA), the accuracy of purely clinical prediction models remains modest. This reinforces the notion that weight gain after ART initiation is a multifactorial process, driven by unmeasured behavioral variables and biological variability that transcend specific national cohorts. Taken together, both studies demonstrate that even in more homogeneous or specialized datasets, the accuracy of prediction remains modest. This reinforces the notion that weight gain after ART initiation is a multifactorial process, partly driven by unmeasured variables and biological variability, which may limit the potential of purely data-driven prediction approaches.

Finally, beyond its immediate results, this work illustrates the broader challenge of applying artificial intelligence to real-world, multicenter clinical cohorts. Machine learning is highly sensitive to missing data, measurement inconsistencies, and unrecorded confounders. While we managed missing data using KNN imputation on variables with <30% missingness, we acknowledge that this method assumes data are missing at random and does not account for the uncertainty of the imputation as robustly as multiple imputation methods might. However, given that the primary predictor (baseline weight) was complete for all subjects, the impact of this limitation on overall performance is likely contained.

## Conclusion

In summary, while machine learning applied to a large, heterogeneous national cohort was able to capture general weight trends, standard baseline clinical variables proved insufficient to provide accurate individual predictions. This result suggests that the limitation lies not within the modeling methodology, but rather in the inherent noise of real-world measurements and the absence of key behavioral determinants in routine Electronic Health Records. Consequently, this study highlights the critical importance of shifting focus from simply increasing sample size to improving data quality and integrating patient-reported outcomes to unlock the full potential of predictive analytics in HIV care.

Legros, G. Mchantaf, C. Mille, Y. Mohamed-Kassim, T. Prazuck, A. Sève, L. Vitry d'Aubigny

## Supporting information

**S1 Table. Predictors selection. (A) Comorbidities, (B) Co-medications and (C) Antiretroviral treatment.** INSTI, Integrase strand transfer inhibitor; TAF, Tenofovir alafenamide; NRTI, Nuclos(t)idic reverse transcriptaseinhibitor; NNRTI, Non nuclos(t)idic reverse transcriptaseinhibitor; PI, Protease inhibitor.
(DOCX)

**S2 Table. Hyperparameters setting for final XGBoost models. The following hyperparameters were tuned using grid search with 10-fold cross-validation via tidymodels package.**
(DOCX)

**S3 Table. Tables from Dat'AIDS database.**
(DOCX)

**S4 Table. Characteristics of the subpopulations of PLHIV treated in the first line used for weight prediction at 6, 12 and 24 months: maximum variation in weight per month limited to 5 kg (A) and 3 kg (B).** Values in number (%), 1Average (quartiles). †Not provided: 537 (8.04), ‡Not provided: 410 (7.23), §Not provided: 285 (6.03). $Not specified: 530 (8.10), $$Not specified: 409 (7.31), $$$Not specified: 284 (6.06).
(DOCX)

**S5 Table. Performance using Weight_T0 = weight at checkpoints. ªValue obtained after 10 cross-validation.** The RMSE evaluates the accuracy of the model by measuring the average difference between the actual values and the predictions (the lower it is, the better the performance of the model). The R2 of RMSE measures the quality of fit of the model in relation to the variability of the real data (0: no fit, 1: perfect fit). The relative RMSE expresses the relative error of the mean compared to the actual values. Relative bias measures the systematic error of the model compared to actual values.
(DOCX)

**S6 Table. Performance of XGBoost models for weight prediction at 6, 12 and 24 months in study subpopulations. (A) Subpopulation limited to 5 kg/month, (B) Subpopulation limited to 3 kg/month.** ªValue obtained after 10 cross-validation. The RMSE evaluates the accuracy of the model by measuring the average difference between the actual values and the predictions (the lower it is, the better the performance of the model). The R2 of RMSE measures the quality of fit of the model in relation to the variability of the real data (0: no fit, 1: perfect fit). The relative RMSE expresses the relative error of the mean compared to the actual values. Relative bias measures the systematic error of the model compared to actual values.
(DOCX)

## Acknowledgments

Members of the Dat'AIDS study group: referent Dr Laurent Hocqueloux (lau <laurent.hocqueloux@chu-orleans.fr)

1.Besançon: C. Chirouze, K. Bouiller, F. Bozon, AS. Brunel, L. Hustache-Mathieu, J. Lagoutte, Q. Lepiller, S. Marty-Quinternet, L. Pépin-Puget, B. Rosolen, N. Tissot, C. Lebreton, L. Bohard

2.Brest: S. Jaffuel, S. Ansart, Y. Quintric, S. Rezig, P. Gazeau, R. Paret, A. Coste, S. Rolland, JC. Duthe

3.Clermont-Ferrand: C. Jacomet, N. Mrozek, C. Theis, M. Vidal, C. Richaud, V. Corbin, A. Benelhadj, A. Zaghdoudi, C. Aumeran, O. Baud, M. Berthommier, M. Charles, C. Durand, D. Coban, A. Mirand, A. Brebion, H. Chabrolles, O. Perruche, E. Creuzet, C. Henquell

4.Guadeloupe: I. Lamaury, G. Baronnet, F. Bissuel, F. Boulard, A. Chéret, J. Coussement, E. Curlier, T. Dequidt, C. Desfontaines, S. Devatine, E. Duvallon, I. Fabre, C. Herrmann-Storck, C. Loraux, S. Markowicz, M. Marquet, R. Ouissa, S. Peugny, L. Pradat-Paz, M. C. Receveur, J. Reltien, K. Samar, K. Schepers, B. Tressieres, V. Walte

5.La Roche sur Yon: D. Merrien, O. Bollangier, D. Boucher, T. Guimard, L. Laine, S. Leautez, M. Morrier, P. Perré

6.La Rochelle: M. Roncato-Saberan, X. Pouget-Abadie, C. Chapuzet, A. Thomas

7.Limoges: JF. Faucher, A. Cypierre, S. Ducroix-Roubertou, H. Durox, C. Genet-Villeger, J. Pascual, P. Pinet, C. Codde, S. Rogez, JB. Woillard, C. Benoist, S. Mafi

8.Lyon: A. Becker, M. Godinot, F. Ader, M. Bonjour, E. Braun, C. Brochier, F. Brunel-Dalmas, P. Chiarello, A. Conrad, S. Degroodt, P. Fascia, T. Ferry, V. Gueripel, V. Icard, J. Izard, C Javaux, H. Lardot, J. Lippmann, D. Makhloufi, Y. Merad, T. Perpoint, S. Roux, S. Sahyouni, M. Simon, S. Soueges, C. Triffault-Fillit, F. Valour, L. Van den Bogaart, M. Wan, AS. Batalla

9.Marseille IHU Méditerrannée: A. Ménard, Y. Belkhir, P. Colson, C. Dhiver, M. Martin-Degioanni, A. Motte, C. Toméi, M. Million, N. De Palmas, M. Champeaux, I. Ravaux

10.Marseille Ste Marguerite: S. Brégigeon, O. Zaegel-Faucher, H. Laroche, MJ. Ducassou, A. Ivanova, I. Jaquet, V. Obry-Roguet, M. Orticoni, E. Ressiot, AS. Ritleng, F. Niemetzky, C. Ferron

11.Martinique: A. Cabié, S. Abel, O. Cabras, L. Cuzin, G. Dos Santos, L. Facelina, L. Fagour, L. de Ghellinck, K. Guit-teaud, E. Louis-Michel, E. Medo, F Quenard, S. Pierre-François, P Richard, A Schapira, B. Tregan

12.Metz: C. Robert, Z. Cavalli, L. Bucy, C. Emilie, A. Fournier

13.Montpellier: A. Makinson, A. Artiaga, M. Bistoquet, E Delaporte, V. Le Moing, J. Lejeune, N. Meftah, C. Merle de Boever, B. Montes, L. Perez, N. Pansu, J. Reynes, C. Tramoni, E. Tuaillon

14.Nancy: B. Lefèvre, M. André, S. Bevilacqua, L. Boyer, MP. Grandin, A. Charmillon, M. Delestan, E. Frentiu, F. Goeh-ringer, S. Hénard, E. Jeanmaire, C. Rabaud, L. Lalevée, J. Kotzyba

15.Nantes: C. Allavena, E. André-Garnier, A. Asquier-Khati, V. Bellon, E. Billaud, C. Biron, B. Bonnet, S. Bouchez, D. Boutoille, J. Brochon, C. Brunet-Cartier, M. Cavellec, L. Collias, C. Deschanvres, T. Drumel, BJ. Gaborit, M. Gregoire, T. Jovelin, R. Lecomte, M. Lefebvre, M. Le Goff, C. Mear-Passard, P. Morineau, C. Moyon, E. Paredes, V. Pineau, G. Querne, A. Soria

16.Nice: D Chirio, P. Pugliese, C. Bonnefoy, M. Buscot, M. Carles, A. Courdurié, J. Courjon, E. Cua, P. Dellamonica, E. Demonchy, A. De Monte, S. Ferrando, C. Pradier, K. Risso, A. Viot, S. Wehrlen-Pugliese

17.Niort: S. Sunder, V. Goudet, A. Dos Santos, V. Rzepecki, A. Metais

18.Orléans: L. Hocqueloux, R. Albert, V. Avettand-Fènoël, S. Bafong-Ketchemen, G. Béraud, J. Effa, C. Gubavu, V.

19.Paris APHP Bicêtre: C. Goujard, A. Castro-Gordon, P. David-Chevallier, V. Godard, Y. Quertainmont, E. Teicher/ S. Jaureguiberry, L. Escaut, B. Henry, C. Couzigou, O. Derradji, R. Collarino, J. Y. Liotier, M. Merad, L. Lévi, L. Lefèvre, R. Courtois

20.Paris APHP Bichat: V. Joly, A Bachelard C. Charpentier, D. Descamps, M. Digumber, A. Gervais, J. Ghosn, Z. Julia, R. Landman, F.Ouvrard, N. Peiffer-Smadja, G. Peytavin, C. Rioux, Y. Yazdanpanah, L. Deconinc

21.Paris APHP Necker/Institut Pasteur: C. Duvivier, K. Amazzough, G. Benabdelmoumen, P. Bossi, G. Cessot, PH. Consigny, M. Garzaro, E. Gomes-Pires, P. Hochedez, O. Itani, K. Jidar, E. Lafont, F. Lanternier, O. Lortholary, C. Louisin, J. Lourenco, C. Melenotte, P. Parize, C. Rouzaud, A. Serris, F. Taieb, J. Zeggagh

22.Paris APHP Pitié Salpetrière: V Pourcher, MA. Valantin, C. Katlama, L. Schneider, S. Seang, R. Tubiana, A. Faycal, S. Saliba, M. Favier, C. Aubron, R. Agher, Y. Dudoit, N. Hamani, N. Qatib, A. Chermak, M. Chansombat, G. Osseni, D. Beniken, A. Nadour

23.Quimper: N. Hall, P. Perfezou, JC. Duthe, FB. Drevillon, JP. Talarmin, L. Khatchatourian, P. Petitgas, P. Martinet

24.Reims: F. Bani-Sadr, V. Brodard, M. Hentzien, I. Kmiec, D. Lambert, D. Lebrun, M. Moutel, M. Petithomme-Nanrocki, A. Brunet, H. Marty, Y. N'Guyen, C. Strady, V. Greigert,

25.Rennes: C. Arvieux, M. Baldeyrou, F. Benezit, G. Bury, M.Cailleaux, JM. Chapplain, M. Dupont, JC. Duthé, S. Ismaël, T. Jovelin, A. Lebot, F. Lemaitre, D. Luque-Paz, A. Maillard, C. Morlat, S. Patrat-Delon, L. Picard, M. Poisson-Vannier, L. Poussier, C. Pronier M. Revest, M. Sebillotte, P. Tattevin, C. Thoreux

26.St Etienne: A. Gagneux-Brunon, E. Botelho-Nevers, A. Pouvaret, F. Saunier, V. Ronat

27.Strasbourg: A. Ursenbach, C. Cheneau, C. Bernard-Henry, S. Fafi-Kremer, P. Gantner, C. Mélounou, P. Klee, Y. Hansmann, N. Lefebvre, Y. Ruch, F. Danion, B. Hoellinger, T. Lemmet, V. Gerber, JM. Schevin, A. Fuchs, C. Le Hyaric, D. Rey

28.Toulouse: P. Delobel, M. Alvarez, N. Biezunski, X. Boumaza, A. Chan Sui Ko, N. Collercandy, A. Debard, C. Delp-ierre, P. Gandia, C. Garnier, R. Gueneau, L. Lelièvre, G. Martin-Blondel, C. Rastoll, S. Raymond, C. Vellas

29.Tourcoing: O. Robineau, E. Aïssi, I. Alcaraz, E. Alidjinou, V. Baclet, A. Boucher, V. Derdour, B. Lafon-Desmurs, A. Meybeck, M. Tetart, M. Valette, N. Viget, A. Diarra, E Bontemps, B Capelliez, P Coulon

30.Vannes: G. Corvaisier, M. Brière, M. De La Chapelle, M. Gousseff, R. Nguyen Van, M. Thierry

## Author contributions

**Conceptualization:** Cyrielle Codde, Jean-François Faucher, Jean-Baptiste Woillard.

**Formal analysis:** Cyrielle Codde, Clément Benoist, Jean-Baptiste Woillard.

**Funding acquisition:** Cyrielle Codde.

**Investigation:** Cyrielle Codde, Jean-Baptiste Woillard.

**Methodology:** Cyrielle Codde, Clément Benoist, Laurent Hocqueloux, Jean-Baptiste Woillard.

**Project administration:** Laurent Hocqueloux, Cyrille Delpierre, Clotilde Allevena, Amélie Ménard, Antoine Chéret, Cédric Arvieux.

**Resources:** Cyrille Delpierre.

**Software:** Clément Benoist.

**Supervision:** Jean-Baptiste Woillard.

**Validation:** Cyrielle Codde.

**Writing – original draft:** Cyrielle Codde, Jean-François Faucher, Jean-Baptiste Woillard.

**Writing – review & editing:** Cyrielle Codde, Jean-François Faucher, Jean-Baptiste Woillard.

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
