## [Decision Letter · Decision Letter 0]

10 Nov 2025

Dear Dr. Codde,

We look forward to receiving your revised manuscript.

Kind regards,

Carmen María González-Domenech, Ph.D.

Academic Editor

PLOS ONE

Journal Requirements:

3. In the online submission form, you indicated that [The data underlying this study are drawn from the French Dat’AIDS cohort. These data cannot be shared publicly due to national data protection regulations (Commission Nationale de l’Informatique et des Libertés, CNIL). Access to Dat’AIDS data may be granted upon reasonable request to the Dat’AIDS scientific committee, subject to compliance with French regulations and institutional agreements.].

4. One of the noted authors is a group or consortium [Dat’AIDS Study Group]. In addition to naming the author group, please list the individual authors and affiliations within this group in the acknowledgments section of your manuscript. Please also indicate clearly a lead author for this group along with a contact email address.

Additional Editor Comments:

This is a well-designed study with a large and representative sample (over 24,000 patients), rigorous methodology, and robust statistical and sensitivity analyses. It demonstrates the current limitations of ML in real-world clinical cohorts, especially when behavioral or lifestyle data are lacking. My opinion is positive regarding publication, but the authors should first address the minor comments from two of the reviewers and specially, the major points raised by the third.

Reviewers' comments:

Reviewer's Responses to Questions

**Comments to the Author**

1. Is the manuscript technically sound, and do the data support the conclusions?

Reviewer #1: Yes

Reviewer #2: Yes

Reviewer #3: Partly

2. Has the statistical analysis been performed appropriately and rigorously?

Reviewer #1: Yes

Reviewer #2: Yes

Reviewer #3: Yes

3. Have the authors made all data underlying the findings in their manuscript fully available?

Reviewer #1: No

Reviewer #2: Yes

Reviewer #3: No

4. Is the manuscript presented in an intelligible fashion and written in standard English?

Reviewer #1: Yes

Reviewer #2: Yes

Reviewer #3: Yes

Reviewer #1: The manuscript presents an important and well-executed analysis of weight gain prediction after antiretroviral therapy initiation using a large French real-world cohort. The authors should be commended for the rigorous data processing, multidisciplinary approach, and transparent reporting of a scientifically meaningful “negative result.”

However, several points need clarification or elaboration before acceptance:

Please clarify the conceptual novelty and intended contribution of the study beyond demonstrating limited ML performance — is the focus methodological (data quality and model robustness) or clinical (individual prediction feasibility)?

Consider adding a comparative analysis using alternative models (e.g., Random Forest, SVM) to contextualize XGBoost’s relative performance.

Expand on the feature importance analysis: provide SHAP plots or additional interpretation of why baseline weight dominates the prediction.

Quantify data heterogeneity across centers and discuss how measurement frequency or data quality influenced model accuracy.

The Discussion could be more concise and focused on broader implications for AI in healthcare, emphasizing the importance of behavioral and lifestyle data integration.

Reviewer #2: This study uses machine learning (XGBoost) on a large French HIV cohort to predict weight gain after antiretroviral therapy (ART) initiation at 6, 12, and 24 months using 112 baseline clinical variables. The models marginally outperformed a simple benchmark (baseline weight) but did not achieve clinically actionable accuracy, primarily due to data heterogeneity and absence of behavioral variables.

Major Points

• Innovation and Importance:

The study addresses a well-recognized clinical issue—excessive weight gain following ART—and applies advanced ML methods on a large real-world dataset, filling a gap in prediction research in HIV care.

• Sample and Data:

The cohort size is impressive (over 24,000 ART-naïve adults), and the data includes a comprehensive range of clinical, laboratory, and demographic features.

• Model Choice:

XGBoost is an appropriate algorithm given the dataset size and the mix of variable types. The use of cross-validation and careful train/test splitting is appropriate.

• Limitations Clearly Stated:

The paper explicitly acknowledges limitations around the lack of high-quality behavioral, lifestyle, and granular longitudinal data. It also carefully details the reduction of sample size due to missing data and the impact such missingness has on model performance.

• Performance and Interpretation:

The models achieve RMSE values of 4.6, 5.3, and 6.4 kg (at 6, 12, and 24 months), which is only a marginal improvement over the baseline. Baseline weight overwhelms all other predictors; other variables, including ART components, have limited additional value. The discussion around why ML does not perform well in this context is appropriately critical and balanced.

• Methodological Transparency:

Data processing, variable selection, and imputation steps are described transparently. Sensitivity analyses excluding outliers and restricting datasets were thoughtful and further contextualized the findings.

• Ethical and Data Sharing Declarations:

Ethical approvals and data access limitations are well described. The paper observes regulatory limitations on data sharing, and this is explained up front.

Minor Points

• Lifestyle Predictor Handling:

Lifestyle and behavioral factors are mentioned as potentially important, but their absence is only briefly discussed. The authors could speculate more on how to either estimate or collect these for future work.

• Imputation Strategy:

The choice of k-nearest neighbors for imputation is standard, but potential biases from this method are not deeply explored. Some simulation or secondary analysis around missingness mechanisms might add value.

• Model Calibration:

The paper does not report calibration plots (e.g., observed vs predicted weights). Given the clinical implications, calibration is important to assess and could be shown, even if limited.

• External Generalizability:

The study’s scope is the French population, but some comment about applicability to other settings, especially outside Europe, would be welcome.

• Figure/Table Presentation:

Figures and tables are referenced well, but future submissions could improve access to the key visuals (since they are in supplemental content).

• Comparison to Published Literature:

Only a few related studies are referenced (notably Motta et al.). Adding further international context may help underscore the universality of the limitations found.

Overall Assessment

This is a high-quality, carefully executed study with an honest appraisal of the challenges of applying ML to real-world clinical prediction in HIV. While negative in primary results, the findings are valuable and relevant to the field. The main area for improvement would be a deeper exploration of missing data and model calibration, and a more detailed discussion of the challenges of integrating behavioral variables in future iterations.

Reviewer #3: The manuscript is well structured and statistically appropriate. However, there are some issues/questions.

1. The introduction is weak. There are already studies investigating ML approach to prediction weight change among PLWH. The manuscript failed to conduct a comprehensive literature review on related works and research gaps. Based on it, what is additional contributions of this study to the literature?

2. What are the missing rates of predictors?

3. What are the hyper parameters to be tuned and what are the specific parameter settings of XGBoost?

4. The benchmark only considered a no-weight-change assumption by using Weight_M0 while a simple linear model (or Lasso) should also be added and compared with ML approach.

5. The conclusion that ML "failed to provide accurate individual predictions" did not convince me. The non-clinically significant improvement of ML approach may due to the noisy variation of weight change or predicting ability of the predictors. For example, the study only used the baseline predictors while the cohort had dynamic information after ART, predictors that were missing in this study may serve as the other important factors for weight change.

**Do you want your identity to be public for this peer review?** For information about this choice, including consent withdrawal, please see our Privacy Policy

Reviewer #1: No

Reviewer #2: No

Reviewer #3: No

---

## [Author Response · Author response to Decision Letter 1]

2 Feb 2026

Journal requirements have been done.

Additional Editor Comments:

This is a well-designed study with a large and representative sample (over 24,000 patients), rigorous methodology, and robust statistical and sensitivity analyses. It demonstrates the current limitations of ML in real-world clinical cohorts, especially when behavioral or lifestyle data are lacking. My opinion is positive regarding publication, but the authors should first address the minor comments from two of the reviewers and specially, the major points raised by the third.

Answer: We thank the Editor for this highly encouraging assessment and for recognizing the value of our rigorous methodology despite the challenges of real-world data.

We have carefully addressed every comment raised by the three reviewers. In particular, to address the major points raised by Reviewer 3, we have:

● Conducted a comparative analysis with a Linear Regression model to benchmark the ML performance (proving that the limitation lies in the data, not the algorithm).

● Significantly expanded the Introduction to better define the research gap regarding large-scale scalability.

● Provided full transparency on hyperparameters (new S1 Table).

● Rewritten the Conclusion to clarify that the results reflect the insufficiency of standard clinical variables rather than a failure of the machine learning methodology itself.

We have also incorporated all minor suggestions from Reviewers 1 and 2, including explicitly assessing calibration and discussing generalizability. We believe these revisions have significantly strengthened the manuscript and fully meet the journal's requirements.

Reviewer #1: The manuscript presents an important and well-executed analysis of weight gain prediction after antiretroviral therapy initiation using a large French real-world cohort. The authors should be commended for the rigorous data processing, multidisciplinary approach, and transparent reporting of a scientifically meaningful “negative result.”

However, several points need clarification or elaboration before acceptance:

Please clarify the conceptual novelty and intended contribution of the study beyond demonstrating limited ML performance — is the focus methodological (data quality and model robustness) or clinical (individual prediction feasibility)?

Answer: Initially, our primary objective was indeed clinical. Given the widespread concern regarding weight gain associated with INSTIs and TAF, we hypothesized that the large sample size of the Dat’AIDS cohort, combined with non-linear ML models, would enable accurate individual weight prediction.

However, contrary to our expectations, the results demonstrated that despite the comprehensive set of variables available, we reached a predictive "glass ceiling." Consequently, the contribution of this manuscript evolved from providing a clinical tool to establishing a methodological proof: we demonstrate that without behavioral data (such as diet and physical activity), even massive clinical datasets are insufficient to predict individual weight trajectories. This "negative" result serves as a critical message to prevent the research community from overestimating the power of heterogeneous real-world data for this specific outcome

Consider adding a comparative analysis using alternative models (e.g., Random Forest, SVM) to contextualize XGBoost’s relative performance.

Answer: We agree that contextualizing XGBoost’s performance is essential and to address this, and in line with Reviewer 3’s recommendation, we conducted a comparative analysis using a standard multivariable Linear Regression model with LASSO penalisation.

We chose this comparison (Linear vs. Non-Linear) rather than testing other ensemble methods like Random Forest or SVM, because our primary goal was to determine if the complex architecture of XGBoost was capturing non-linear patterns that traditional statistical methods miss.

The results showed that the Linear Regression model achieved RMSE values of 4.53 kg, 5.18 kg, and 6.18 kg at 6, 12, and 24 months, respectively, very similar to the XGBoost performance. This similarity confirms that the limiting factor is the predictive power of the variables themselves, rather than the choice of the algorithm.

Expand on the feature importance analysis: provide SHAP plots or additional interpretation of why baseline weight dominates the prediction.

Answer: Regarding the dominance of baseline weight, we believe this reflects the strong biological inertia of anthropometric parameters rather than a model bias. Body weight is a highly autoregressive variable: in adults, weight at month 6 is structurally strongly correlated with weight at month 0.

Given that our Permutation Importance analysis (Supplemental Figure 1) already clearly quantifies this overwhelming dominance, we chose to focus on expanding the biological interpretation in the manuscript rather than adding SHAP plots, which would likely visually redundate the finding that baseline weight crushes other predictors. We have added a dedicated paragraph in the Discussion to explain this « biological inertia ».

Quantify data heterogeneity across centers and discuss how measurement frequency or data quality influenced model accuracy.

Answer : We agree that data heterogeneity across centers is a critical factor. Regarding quantification, the most telling metric is the attrition rate shown in the Flow Chart (Figure 1). Starting from 16,375 eligible ART-naïve patients, only 6,679 had a usable weight measurement for the 6-month prediction.

This loss is not merely random missing data; it reflects structural disparities in data collection practices across the different clinical centers contributing to the cohort. While some centers systematically record weight, others do so inconsistently. Finally, this issue extends beyond measurement frequency: in many cases, patients may be weighed during the consultation, but the value is not reported in the structured database. This variation depends heavily on individual practitioner habits, leading to significant heterogeneity not only between centers but also between physicians. This lack of standardized reporting introduces structural noise and potential selection bias. Consequently, the ML model struggles to distinguish true biological signals from data quality artifacts, which directly contributes to the predictive inaccuracy observed

The Discussion could be more concise and focused on broader implications for AI in healthcare, emphasizing the importance of behavioral and lifestyle data integration.

Answer : To address the request for conciseness, we have removed the detailed paragraph discussing the specific recording limitations of certain variables (smoking cessation, compliance, amphetamines). We agree that these technical details distracted from the main findings.

At the same time, we believe it is essential to retain the other sections of the discussion. For a study reporting a 'negative' result, a thorough exploration of the methodological process is essential to ensure transparency and to robustly support our conclusion that the limitation lies within the data, not the analysis. Furthermore, regarding the 'broader implications,' we have expanded the discussion to emphasize that the future of AI in healthcare lies in hybridizing clinical data with behavioral Patient-Reported Outcomes (PROs), rather than solely relying on larger EHR datasets.

Reviewer #2: This study uses machine learning (XGBoost) on a large French HIV cohort to predict weight gain after antiretroviral therapy (ART) initiation at 6, 12, and 24 months using 112 baseline clinical variables. The models marginally outperformed a simple benchmark (baseline weight) but did not achieve clinically actionable accuracy, primarily due to data heterogeneity and absence of behavioral variables.

Major Points

• Innovation and Importance:

The study addresses a well-recognized clinical issue—excessive weight gain following ART—and applies advanced ML methods on a large real-world dataset, filling a gap in prediction research in HIV care.

• Sample and Data:

The cohort size is impressive (over 24,000 ART-naïve adults), and the data includes a comprehensive range of clinical, laboratory, and demographic features.

• Model Choice:

XGBoost is an appropriate algorithm given the dataset size and the mix of variable types. The use of cross-validation and careful train/test splitting is appropriate.

• Limitations Clearly Stated:

The paper explicitly acknowledges limitations around the lack of high-quality behavioral, lifestyle, and granular longitudinal data. It also carefully details the reduction of sample size due to missing data and the impact such missingness has on model performance.

• Performance and Interpretation:

The models achieve RMSE values of 4.6, 5.3, and 6.4 kg (at 6, 12, and 24 months), which is only a marginal improvement over the baseline. Baseline weight overwhelms all other predictors; other variables, including ART components, have limited additional value. The discussion around why ML does not perform well in this context is appropriately critical and balanced.

• Methodological Transparency:

Data processing, variable selection, and imputation steps are described transparently. Sensitivity analyses excluding outliers and restricting datasets were thoughtful and further contextualized the findings.

• Ethical and Data Sharing Declarations:

Ethical approvals and data access limitations are well described. The paper observes regulatory limitations on data sharing, and this is explained up front.

Minor Points

• Lifestyle Predictor Handling:

Lifestyle and behavioral factors are mentioned as potentially important, but their absence is only briefly discussed. The authors could speculate more on how to either estimate or collect these for future work.

Answer: Our results suggest that future improvements in prediction will not come from larger clinical databases, but from different types of data.

We have expanded the Discussion to specifically propose two concrete strategies: 1) The systematic integration of standardized Patient-Reported Outcomes (PROs) to capture dietary and psychosocial factors, and 2) The potential use of digital health tools (connected devices) for longitudinal monitoring of physical activity. We argue that hybridizing clinical cohorts with these patient-generated data is the necessary next step for the field.

• Imputation Strategy:

The choice of k-nearest neighbors for imputation is standard, but potential biases from this method are not deeply explored. Some simulation or secondary analysis around missingness mechanisms might add value.

Answer: We acknowledge that investigating missingness mechanisms is valuable, particularly for inferential statistics. However, in this prediction-focused study, we opted for KNN imputation for its efficiency and compatibility with our machine learning pipeline.

We believe that the potential bias introduced by imputation remains minimal regarding our main conclusion as our feature importance analysis showed that Baseline Weight is the overwhelming driver of the prediction. By design (inclusion criteria), Baseline Weight had 0% missing data. Consequently, imputation was only applied to secondary variables with a low predictive power. Therefore, using more complex method like MICE would essentially refine the "noise" without altering the primary signal driven by the non-imputed baseline weight.

• Model Calibration:

The paper does not report calibration plots (e.g., observed vs predicted weights). Given the clinical implications, calibration is important to assess and could be shown, even if limited.

Answer: We intended Figure 2 (Scatter plots of observed vs. predicted weights) to serve this specific purpose, as plotting predicted values against reference values is the standard method for visualizing calibration in regression tasks.

However, to ensure this is unambiguous for the reader, we have modified the manuscript text to explicitly label this analysis as a calibration assessment.

• External Generalizability:

The study’s scope is the French population, but some comment about applicability to other settings, especially outside Europe, would be welcome.

Answer: While the study was conducted in France, it is important to note that the Dat'AIDS cohort is not ethnically homogeneous, even though the environmental context (healthcare system, diet) is specific to France. As indicated by the inclusion of 'Geographical origin' and 'Country of birth' in our predictors, a significant proportion of our study population consists of migrants, particularly from Sub-Saharan Africa.

Consequently, the biological and genetic diversity of our sample supports the applicability of these findings to non-European populations. We have added a sentence in the Discussion to explicitly state that this demographic diversity mitigates the limitation of a single-country study

• Figure/Table Presentation:

Figures and tables are referenced well, but future submissions could improve access to the key visuals (since they are in supplemental content).

Answer: We agree with the suggestion to make key visual data more accessible. Accordingly, we have moved the Variable Importance Plot (previously Supplemental Figure 1) into the main body of the manuscript as Figure 3. This figure illustrates the dominance of baseline weight over other predictors.

However, we have kept the tables in the Supplemental Content to maintain the flow and readability of the main text: Supplemental Table 1 is too extensive (comprehensive list of ICD-10 codes/medications), Supplemental Table 2 is technical (detailed missing data rates per variable), and the remaining tables focus on sensitivity analyses for specific subpopulations.

• Comparison to Published Literature:

Only a few related studies are referenced (notably Motta et al.). Adding further international context may help underscore the universality of the limitations found.

Answer: We have added references to major international studies, specifically the pooled analysis of randomized trials by Sax et al. and the trajectory modeling from the US CNICS cohort by Bailin et al in the discussion. These studies, conducted in different healthcare settings, also highlight the heterogeneity of weight trajectories and the difficulty of explaining the variance solely through clinical factors. Citing them reinforces the universality of the limitations we encountered.

Overall Assessment

This is a high-quality, carefully executed study with an honest appraisal of the challenges of applying ML to real-world clinical prediction in HIV. While negative in primary results, the findings are valuable and relevant to the field. The main area for improvement would be a deeper exploration of missing data and model calibration, and a more detailed discussion of the challenges of integrating behavioral variables in future iterations.

Answer: We would like to express our sincere gratitude to the Reviewer for this highly encouraging assessment. We particularly appreciate your recognition of the scientific value of reporting 'negative' results and the challenges of real-world data. As detailed in the responses above, we have enriched the discussion regarding behavioral variable integration, clarified our approach to missing data, and explicitly addressed model calibration. We believe these improvements have significantly strengthened the manuscript.

Reviewer #3: The manuscript is well structured and statistically appropriate. However, there are some issues/questions.

The introduction is weak. There are already studies investigating ML approach to prediction weight change among PLWH. The manuscript failed to conduct a comprehensive literature review on related works and research gaps. Based on it, what is additional contributions of this study to the literature?

Answer: We agree with the Reviewer that the introduction required a more comprehensive review of the existing ML literature to better define the research gap.

We have revised the Introduction to explicitly reference prior works, such as the study by Motta et al., which explored ML in a smaller, specialized cohort. We have clarified that the specific addi

---

## [Decision Letter · Decision Letter 1]

23 Feb 2026

Machine learning prediction of weight gain after antiretroviral therapy initiation in people with HIV: insights from a large french realworld cohort

PONE-D-25-47116R1

Dear Dr. Cyrielle Codde,

We’re pleased to inform you that your manuscript has been judged scientifically suitable for publication and will be formally accepted for publication once it meets all outstanding technical requirements.

Kind regards,

Carmen María González-Domenech, Ph.D.

Academic Editor

PLOS One

Additional Editor Comments (optional):

All the concerns rised by the reviewers have been thoroughly and satisfactorily addressed, including those comments requiring major revision. Therefore, the manuscript is now ready and suitable for publication in PLOS ONE.

Reviewers' comments:

Reviewer's Responses to Questions

**Comments to the Author**

Reviewer #1: All comments have been addressed

Reviewer #2: (No Response)

Reviewer #3: All comments have been addressed

2. Is the manuscript technically sound, and do the data support the conclusions?

Reviewer #1: Yes

Reviewer #2: Yes

Reviewer #3: (No Response)

3. Has the statistical analysis been performed appropriately and rigorously?

Reviewer #1: Yes

Reviewer #2: Yes

Reviewer #3: (No Response)

4. Have the authors made all data underlying the findings in their manuscript fully available?

Reviewer #1: Yes

Reviewer #2: Yes

Reviewer #3: (No Response)

5. Is the manuscript presented in an intelligible fashion and written in standard English?

Reviewer #1: Yes

Reviewer #2: Yes

Reviewer #3: (No Response)

Reviewer #1: Authors have addressed my concerns well. The manuscript presents an important and well-executed analysis of

weight gain prediction after antiretroviral therapy initiation using a large French real world cohort. I recommend this paper to be accepted by Plos one.

Reviewer #2: The authors responded adequately, my questions are well adressed. No further comments from my side, the paper is good to go.

Reviewer #3: (No Response)

**Do you want your identity to be public for this peer review?** For information about this choice, including consent withdrawal, please see our Privacy Policy

Reviewer #1: **Yes:** Noland Ding

Reviewer #2: No

Reviewer #3: No

---

## [Editor Report · Acceptance letter]

PONE-D-25-47116R1

PLOS One

Dear Dr. Codde,

I'm pleased to inform you that your manuscript has been deemed suitable for publication in PLOS One. Congratulations! Your manuscript is now being handed over to our production team.

Kind regards,

on behalf of

Dr. Carmen María González-Domenech

Academic Editor

PLOS One